# Outcomes after Flow Diverter Treatment in Subarachnoid Hemorrhage: A Meta-Analysis and Development of a Clinical Prediction Model (OUTFLOW)

**DOI:** 10.3390/brainsci12030394

**Published:** 2022-03-15

**Authors:** Michelle F. M. ten Brinck, Viktoria E. Shimanskaya, René Aquarius, Ronald H. M. A. Bartels, Frederick J. A. Meijer, Petra C. Koopmans, Guido de Jong, Ajay K. Wakhloo, Joost de Vries, Hieronymus D. Boogaarts

**Affiliations:** 1Department of Neurosurgery, Radboud University Medical Center, 6525 GA Nijmegen, The Netherlands; michelle.tenbrinck@radboudumc.nl (M.F.M.t.B.); vika.shimanskaya@radboudumc.nl (V.E.S.); rene.aquarius@radboudumc.nl (R.A.); ronald.bartels@radboudumc.nl (R.H.M.A.B.); guido.dejong@radboudumc.nl (G.d.J.); joost.devries@radboudumc.nl (J.d.V.); 2Department of Medical Imaging, Radboud University Medical Center, 6525 GA Nijmegen, The Netherlands; anton.meijer@radboudumc.nl; 3Department for Health Evidence, Radboud University Medical Center, 6525 GA Nijmegen, The Netherlands; petra.koopmans@radboudumc.nl; 4Department of Neurointerventional Radiology, Beth Israel Lahey Health, Tufts University School of Medicine, Boston, MA 02111, USA; ajay.wakhloo@lahey.org

**Keywords:** endovascular techniques, flow diverter, intracranial aneurysm, posterior communication artery

## Abstract

Background: patients with a subarachnoid hemorrhage (SAH) might need a flow diverter (FD) placement for complex acutely ruptured intracranial aneurysms (IAs). We conducted a meta-analysis and developed a prediction model to estimate the favorable clinical outcome after the FD treatment in acutely ruptured IAs. Methods: a systematic literature search was performed from 2010 to January 2021 in PubMed and Embase databases. Studies with more than five patients treated with FDs within fifteen days were included. In total, 1157 studies were identified. The primary outcome measure was the favorable clinical outcome (mRS 0–2). Secondary outcome measures were complete occlusion rates, aneurysm rebleeding, permanent neurologic deficit caused by procedure-related complications, and all-cause mortality. A prediction model was constructed using individual patient-level data. Results: 26 retrospective studies with 357 patients and 368 aneurysms were included. The pooled rates of the favorable clinical outcome, mortality, and complete aneurysm occlusion were 73.7% (95% CI 64.7–81.0), 17.1% (95% CI 13.3–21.8), and 85.6% (95% CI 80.4–89.6), respectively. Rebleeding occurred in 3% of aneurysms (11/368). The c-statistic of the final model was 0.83 (95% CI 0.76–0.89). All the studies provided a very low quality of evidence. Conclusions: FD treatment can be considered for complex ruptured IAs. Despite high complication rates, the pooled clinical outcomes seem favorable. The prediction model needs to be validated by larger prospective studies before clinical application.

## 1. Introduction

There are an increasing number of studies which report on the off-label use of FDs in the initial treatment of an acute aneurysmal subarachnoid hemorrhage (SAH) [1].

The application of FDs in the acute phase is much debated due to its potential disadvantages. Delayed aneurysm occlusion associated with the use of an FD could expose the aneurysm to re-rupture. Furthermore, the need for dual antiplatelet therapy (DAPT) in the acute phase may increase the risk of hemorrhagic complications, especially in patients requiring a ventriculostomy. Finally, the prothrombotic phase in SAH could lead to a high rate of thromboembolic complications when intra-luminal devices are used. As conflicting reports have been published about complication rates in early and delayed FD treatments, these concerns have led to extensive discussions about the timing of FD treatment in acute SAH [2,3].

Several previous meta-analyses have addressed this topic [2,3,4,5,6]. Some of these reviews included a limited number of studies with small cohorts, often with three or less patients. In the past years, larger patient populations were studied, which could provide new and more accurate data [7,8]. 

Predictive models for the prediction of outcomes after aneurysmal SAH, regardless of treatment modality, have been published [9,10]. However, these models are based on studies from a time when FD treatment was not an option or was uncommon.

The aim of this study is to provide an up-to-date meta-analysis on clinical outcomes after FD treatment in the setting of acute SAH and to develop a prediction model based on this literature.

## 2. Materials and Methods

### 2.1. Search Strategy and Article Selection

A systematic review of the literature on studies reporting both the clinical and angiographic outcome of SAH patients treated with FDs was performed. The search, study selection, and data extraction process were performed according to the PRISMA guidelines. The search was last performed on 31 December 2020 in PubMed and Embase databases. Studies conducted prior to 2010 were excluded. For the full search strategy, see Appendix B.

Title and abstract screening were performed by two authors (MtB and HB), and disagreements were set by a third reader (VES). Full text screening was performed by two authors (MtB and VES). Inclusion criteria were defined as: articles reporting on both the clinical and radiological outcome of patients treated with any type of flow diverters in the acute phase, defined as within fifteen days following the aneurysm rupture. Fifteen days was set as the cut-off point to improve comparability with existing literature, and since rebleeding and delayed cerebral ischemia (DCI) mostly occur within this timeframe [11]. Therefore, the use of FDs within this period is of significant interest.

Both patients with a first SAH, as well as patients with aneurysm rebleeding (from both untreated and previously treated IAs), were included. Aneurysms treated with flow diversion, plus additional coiling, were also included. Studies were required to report five or more patients.

Studies providing information on both the cohort- and patient-level were included. Patient-level data were retrieved from all the studies that provided this.

For exclusion criteria, see Appendix A. Studies were also checked for duplication in the study population. Studies were excluded if they reported less than five ‘new’ patients.

Corresponding authors were contacted by e-mail if unclear or missing data were crucial for either inclusion or exclusion. In the case of no reply within 2 weeks, the study was excluded.

### 2.2. Outcomes

The primary outcome measure was the favorable clinical outcome, defined by either the modified Rankin Scale (mRS), score 0–2, or the Glasgow Outcome Scale (GOS), score 4–5, at the last available moment of follow-up. Secondary outcome measures were the complete occlusion rates at the last available follow-up imaging, aneurysm rebleeding, permanent neurologic deficit caused by treatment-related complications, and all-cause mortality. Complete occlusion was defined as either Raymond–Roy (RR) class 1 measured on the RR scale or O’Kelly–Marotta (OKM) class D, measured on the OKM scale. A description of complete occlusion in case studies was also allowed. Treatment-related complications were categorized by the type (ischemic/intracranial hemorrhagic/other, e.g., vessel dissection) and timing (intra- or early post-procedural (≤30 days), and late post-procedural (>30 days)).

### 2.3. Data Collection

All pre-specified data items were extracted from publications by two authors (MtB and VES), according to the PRISMA statement, in a pre-specified form. For registered baseline and outcome parameters, see Appendix C.

### 2.4. Bias Evaluation

The GRADE criteria were used to assess the quality of each study [12]. A funnel plot was constructed to detect any possible publication bias.

### 2.5. Statistics

Pooled event rates, including a 95% confidence interval, were calculated with a logit transformation, a DerSimonian–Laird estimator for tau^2^, and an inverse variance method. For individual studies, the Clopper–Pearson confidence interval was calculated. A continuity correction of 0.5 was used in studies with zero cell frequencies. The pooling was performed using package ‘meta’ in R version 4.0.4. For the prediction model developed based on the patient-level dataset, information was available for 100% of the outcome measures (mRS), and for 77–100% of the potential predictors. The cases with missing data were not included in the analysis. A sensitivity analysis was conducted by performing a single imputation of the missing data, using all the predictors from the pooled dataset. For the development of the prediction model, cubic spline functions were used to explore whether continuous variables required transformation. The optimal set of predictors was chosen by univariate analysis combined with clinical judgement. Afterwards, the discrimination and calibration of the model were assessed. The internal validation process was performed by means of bootstrapping with 250 bootstrap replications, in which a shrinkage factor and c-statistic were estimated. Afterwards, a new intercept was calculated with the offset procedure. The prediction model was developed in accordance with the TRIPOD statement [13]. All analyses for the prediction model were performed with IBM SPSS Statistics (version 25.0.0.1) and R Studio 1.1.463.

## 3. Results

### 3.1. Study Population and Patient Characteristics

Twenty-six retrospective case series assessing clinical and imaging outcomes, after FD treatment in the setting of acute SAH, were included (Appendix A) [7,8,14,15,16,17,18,19,20,21,22,23,24,25,26,27,28,29,30,31,32,33,34,35,36,37]. The total population consisted of 357 patients with 368 ruptured aneurysms (Table 1). Patient-level data were retrieved for 300 patients (24 studies). The timing of treatment within 15 days after SAH was confirmed by the corresponding authors in two studies in which this was unclear (Appendix A) [23,27]. Four studies had a duplication in population [7,26,27,33]. Duplicate patients were excluded. Baseline characteristics varied substantially among the studies due to the different inclusion criteria. Variation was observed in the types of aneurysms (several types [14,16,19,22,23,25,26,28,29,30,32,33,36] versus a single type of aneurysm [7,15,17,18,20,21,24,27,31,34,35,37]), location (anterior versus posterior), clinical presentation, and the timing of treatment. For example, of patients with known World Federation of Neurosurgical Societies (WFNS) or Hunt and Hess (HH) grades at presentation (*n* = 348), 96 patients (28%, range 0–71%) showed an unfavorable WFNS or HH grade of 4 or 5. In some series, the patients were treated within twelve hours after admission [31], while in the others, the treatment timing ranged between 0 and 15 days [8,19]. Several studies focused solely on either anterior or posterior circulation aneurysms [7,16,18,21,24,31,35,36,37].

The use of adjunctive coiling was reported in 56/307 aneurysms (18%). In two studies, the number of patients who had coiling was unclear [7,19]. The majority of the aneurysms were blood-blister-like (*n* = 161, 44%), followed by dissecting (*n* = 90, 24%), saccular (*n* = 81, 22%), fusiform (*n* = 32, 9%), pseudoaneurysm (*n* = 3, 1%), and mycotic (*n* = 1, 0.2%). The majority of the aneurysms were located in posterior circulation (*n* = 235, 64%). See Appendix A for the used types and numbers of flow diverters. The used (periprocedural) antiplatelet regimens were often not available for patient-level, and they were significantly heterogeneous between studies. Additionally, data on PRU testing were only sparsely available.

### 3.2. Primary and Secondary Outcomes

A meta-analysis was performed for the favorable clinical outcome, mortality rate, and complete occlusion rate. The rate of the favorable clinical outcome ranged between 21% and 100% (Appendix A). The mean clinical follow-up time varied between 3 and 33 months. 

Based on available data from 25 studies (338 patients), the pooled favorable clinical outcome rate equaled 73.7% (95% CI 64.7–81.0 (Table 1 and Figure 1)). Based on the complete available patient-level data (20 studies, 243 patients), the rates for the favorable clinical outcome for patients treated within 72 h, versus those treated at day 3–15, were 64.0% (87/136), and 80.4% (86/107), respectively, *p* = 0.005. The rates of unfavorable presentation (WFNS or HH 4–5) for these subgroups were 33.6% (45/134), and 15.0% (15/100) (19 studies, 234 patients), *p* = 0.001.

Both the dichotomized treatment delay (within 72 h or ≥72 h) and treatment delay as continuous variables had no statistically significant correlation with the favorable outcome in the multiple logistic regression analysis (Table 2).

The pooled mortality rate (357 patients) equaled 17.1% (95% CI 13.3–21.8 (Figure 2)), varying between 0% and 50%. The mortality rate seemed to be slightly higher in patients with a posterior circulation aneurysm (17% versus 10%), although this difference was not statistically significant in the univariate analysis (*p* = 0.076).

Based on 25 studies (274 patients), the pooled rate of complete aneurysm occlusion equaled 85.6% (95% CI 80.4–89.6 (Figure 3)) at the last available moment of follow-up. This rate was based only on the available data of survivors.

Most studies used a global description of the angiographic outcome, without the use of the RR or OKM scales. Mean/median angiographic follow-ups varied between 3 and 24 months.

The reporting of complications was heterogeneous across the studies. Some reported all types of complications, regardless of their clinical significance [19], while others only reported on hemorrhagic and thrombo-embolic complications [34]. Overall, complications occurred in 24% of patients (87/357, range 0–71% (Appendix A)). In-stent thrombosis and thrombo-embolic complications were the most common procedural complications (*n* = 16), followed by technical complications, such as poor device opening or stent migration (*n* = 7), vessel dissection or perforation (*n* = 5), or ischemia due to branch vessel occlusion (*n* = 2). Two patients experienced perforations due to adjunctive coiling.

Rebleed post-treatment occurred in 3% of the aneurysms (11/368). Most of these cases were treated by FD alone, without adjunctive coiling (9/11, 82%). The majority of aneurysms that rebled were located in the anterior circulation (8/11, 73%), with a mean aneurysm size of 14 mm (range 2–34 mm). These aneurysms were saccular (*n* = 5), fusiform (*n* = 3), and blood-blister-like (*n* = 3). The mortality rate in this group was 55%.

In five studies, it was not reported whether complications led to a permanent neurological deficit [7,14,16,20,25]. A permanent neurological deficit due to a complication was reported in 32/265 patients (12%, range 0–33%).

### 3.3. Quality of Evidence

All but two studies demonstrated a very low quality of evidence. Two studies were graded as showing a low quality of evidence due to their sample size [7,8]. For the GRADE classification of each study, see Appendix A. The funnel plot did not indicate a publication bias (Appendix A).

### 3.4. Prediction Model

Patient-level data were complete with respect to all parameters for 175 patients. Table 2 presents the results from multiple logistic regressions for the favorable mRS (0–2), subsequent to FD treatment in the acute phase after SAH, performed in this cohort. The unfavorable WFNS and HH grades at presentation, aneurysm configuration, aneurysm location and size, and treatment delay (days) were eventually included in the model. After shrinkage of the coefficients, following the bootstrapping procedure and the re-estimation of the intercept, the c-statistic of the final model was 0.83 (95% CI 0.76–0.89). The shrinkage factor was estimated to be 0.89. The Hosmer–Lemeshow test was not significant (*p* = 0.20). As a sensitivity analysis, all analyses were repeated with the imputed data (290 patients), which provided similar results. The model was made available online: www.outflow-tool.com (accessed on 7 February 2022).

## 4. Discussion

In this study, we summarized the outcomes of treatment with FDs for ruptured IAs within a maximum of fifteen days after SAH, and we constructed a clinical prediction model using the individual patient data.

FDs are used for a variety of complex intracranial aneurysms that are not amenable to standard endovascular or open surgical treatment. In the setting of acute SAH, it concerns off-label use.

This is the largest individual patient-level data analysis of FD use in SAH patients to date. It revealed a lower favorable clinical outcome than the two first published meta-analyses on this topic (73% compared to 81–83%), but it was similar to the rate of 72% reported in a meta-analysis by Dossani et al., who also excluded series with fewer than five patients [2,3,4]. These early meta-analyses might be biased by smaller studies. Smaller studies tend to report more positive outcomes, a trend demonstrated by our current meta-analysis (Figure 1 and Figure 2).

Cagnazzo et al. reported an overall complication rate of 17.8%, with a rate of 9% for complications associated with transient and permanent morbidity [2]. Dossani et al. reported an overall complication rate of 27.5% and a mortality rate of 15.5% [3]. Another meta-analysis, that of Foreman et al., reported a rate of only symptomatic neurologic complications of 16.5% [6]. They reported three aneurysm re-ruptures after FD placement, but also three deaths caused by aneurysm rebleeding prior to FD treatment. These deaths were possibly treatment-related, since the antiplatelet use may have been influential. Foreman et al. found that unfavorable presentation was associated with complications. Compared to their study, our population consisted of a larger part with unfavorable presentation (WFNS or HH 4–5): 16% versus 28%. This can contribute to the observed difference of, respectively, 80% and 73%, as the rate of the favorable clinical outcome (mRS 0–2). 

The rebleed rate after treatment was low (3%) and comparable to rates after standard coiling [38]. However, caution must be taken since most studies are retrospective case series, and it may be possible that SAH patients treated with FDs who rebled early were not always included. Post-treatment rebleeding was present in larger aneurysms, and in most cases, no adjunctive coiling was performed. For large aneurysms, coiling could be considered in addition to FD placement.

Although the reporting of complications is very inconsistent across studies, they seem to occur relatively frequently (24%), and a permanent neurological deficit was present in 12% of cases due to complications. These results are comparable to those of stent-assisted coiling [39]. With respect to future research, we propose to report the complication rates of treatment-related complications associated with both transient and permanent morbidity.

Angiographic results regarding complete occlusion (86%) were similar to previously published literature (87.5–90.2%) [3,6].

Comparing the mortality rate of patients treated for acute SAH can be difficult due to differences in patient and aneurysm population. Furthermore, due to technical reasons, it may not be feasible to use a certain treatment modality.

A recent meta-analysis including approximately 1600 patients treated with stent-assisted coiling for SAH found a favorable outcome of 75%, and 57% of the aneurysms were completely occluded at follow-up [40]. The rate of the favorable clinical outcome is comparable to the outcome in this study. Our complete occlusion rate (86%) was higher. This was possibly related to the flow diversion effect.

For approximately 20,000 Canadian patients who were diagnosed with SAH between 2004 and 2015, the in-hospital mortality rate was 22% [40]. This aligns with the all-cause mortality rate of 17% (after follow-up) found in this study. It must be kept in mind that this study consisted of a subgroup of patients with aneurysms that were not amenable to standard treatment options. Therefore, a direct comparison of the mortality rate found in this study with that of general SAH patients would be biased.

### 4.1. Prediction Model

Estimating the probability of the favorable clinical outcome can be difficult, since as shown above, the rates of (clinically significant) complications and the favorable clinical outcome vary widely between published studies. Moreover, the quality of available studies is generally low.

We developed a prediction model (OUTFLOW; OUTcome in FLOWdiverter treatment after SAH) based on the patient-level data from available literature in order to estimate the probability of a good favorable outcome. The model is easy to use as it consists of only five parameters which are routinely collected in clinical practice. It must be noted that patient-specific characteristics, such as age and comorbidities, should always be taken into consideration. Additionally, the quality of available literature is low, and only half of patients included in the meta-analysis could be used for development of the prediction model. Therefore, external validation by high-quality (prospective) studies is required prior to its application in clinical practice.

Our meta-analysis confirmed that unfavorable presentation (WFNS and HH 4–5) is associated with an unfavorable clinical outcome (Table 2), and that aneurysm size is an important factor associated with re-rupture and outcome. Both observations correspond with previous reviews [3,4,6]. Two of these studies used cut-offs for aneurysm sizes of 7 mm and 20 mm that were associated with a poor outcome [3,4]. In our proposed model, continuous variables are used to estimate the outcome.

Aneurysms in the posterior circulation seem to be associated with a higher complication risk or mortality compared to aneurysms in the anterior circulation [41]. The mortality rate in our series seemed to be slightly higher in patients with a posterior circulation aneurysm (17% versus 10%), although the difference was not statistically significant in univariate analysis (*p* = 0.076).

Treatment delay is often a topic of debate. We found no significant relation between treatment delay and clinical outcome. Two other recent meta-analyses also found no significant difference/relation between the treatment time and the (symptomatic) complication rate [3,6]. However, Dossani et al. reported a non-statistically significant slightly greater risk for stroke/death and hemorrhagic complications in the early treatment group.

We observed a non-statistically significant trend for better favorable outcome rates with later treatment, when treatment delay was dichotomized with 72 h as cut-off point. It remains unclear whether early treatment itself increases the risk of complications and an unfavorable outcome. Certain aneurysm characteristics, such as a large size or aneurysm morphology, may confound the decision on treatment timing. External validation of the model should provide more clarity. Early treatment should not be withheld if logistically and technically possible.

### 4.2. Limitations

This study has several limitations. First, the included studies were rated as having a very low quality of evidence. Additionally, the rates of the unfavorable clinical presentation (0–64%) and the favorable clinical outcome (21–100%) varied widely, which could indicate selection bias, despite the symmetrical funnel plot. Furthermore, data with different follow-up periods were used for pooling. To minimize the risk of publication bias, we chose that a minimum of five patients should be included in each study. Second, clear criteria for patient selection and time to treatment were missing from some studies. During the earlier years of the inclusion period, FD treatment was less common or considered a last resort. The threshold to treat complex acute SAH patients may have lowered, due to justification via early case series. This results in a heterogeneous population. In addition, increasing the experience of interventionalists with the use of FDs has likely contributed to a technically safer procedure. Third, the prediction model relies on data with the aforementioned shortcomings. The missing data patterns were variable, as some studies did not report precise treatment delay, aneurysm size, WFNS, or HH. Therefore, the complete case analysis was preferred, and the imputed dataset was used for sensitivity analysis. Finally, this model was validated only internally; the online availability will make future external validation possible to strengthen the model.

## 5. Conclusions

Flow diversion can be used in a selected subgroup of complex, recently ruptured intracranial aneurysms, with satisfactory clinical outcomes. Despite high rates of complications, the pooled clinical outcome is still favorable in 73% of cases based on an up-to-date meta-analysis. The OUTFLOW prediction model to predict the favorable clinical outcome requires further development.

## Figures and Tables

**Figure 1 brainsci-12-00394-f001:**
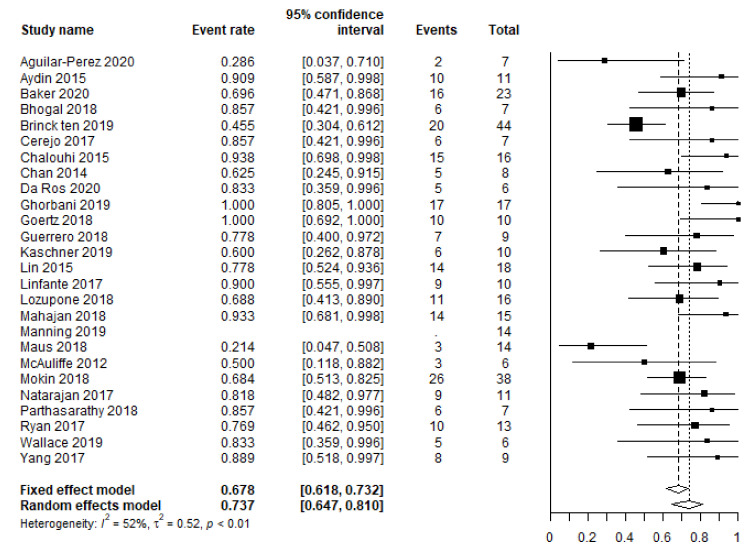
Meta-analysis of the favorable clinical outcome rate.

**Figure 2 brainsci-12-00394-f002:**
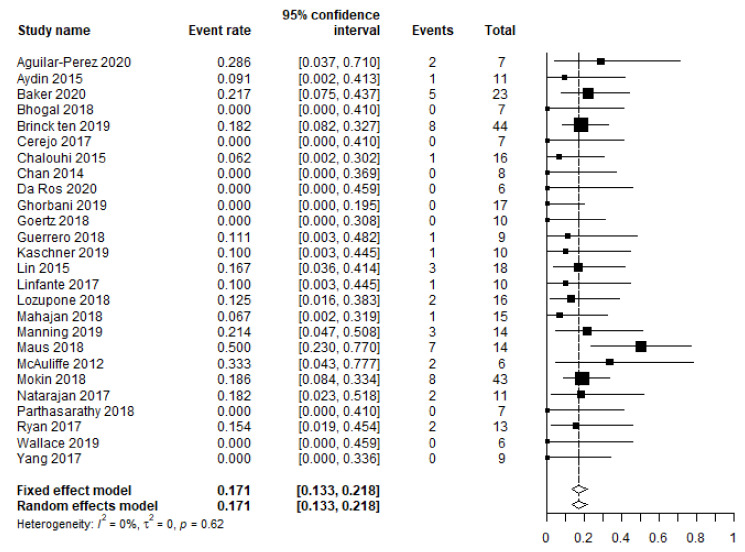
Meta-analysis of all-cause mortality.

**Figure 3 brainsci-12-00394-f003:**
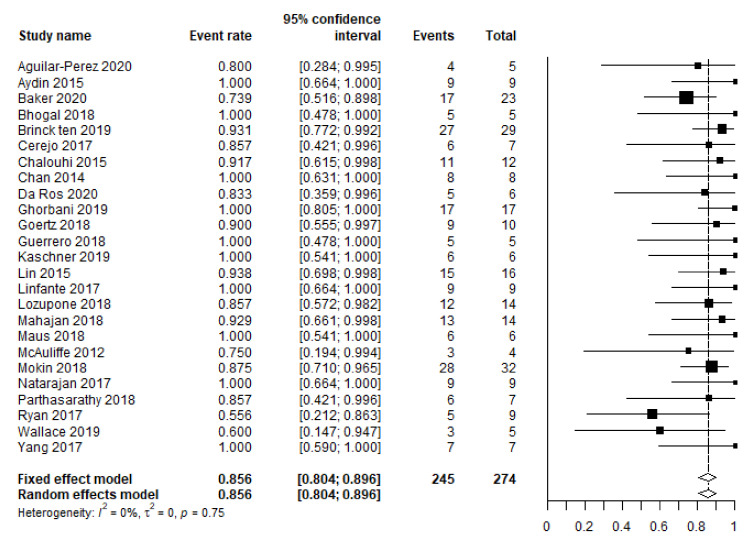
Meta-analysis of the complete occlusion rate.

**Table 1 brainsci-12-00394-t001:** Pooled study baseline characteristics and outcomes.

Pooled Variables	Number (%)	Number of Articles
Total population		
N. eligible patients	357	26
N. eligible aneurysms	368	26
Proportion of unfavorable HH/WFNS grades at presentation	96/348 (28)	25 ^a^
		26
Aneurysm type		
Blood blister-like	161 (44)	
Saccular	81 (22)	
Fusiform	32 (9)	
Dissecting	90 (24)	
Pseudoaneurysm	3 (1)	
Mycotic	1 (0.2)	
Aneurysms located in posterior circulation	235/368 (64)	26
Aneurysms additionally coiled	56/307 (18)	24 ^b^
Favorable clinical outcome (mRS 0–2, GOS 4–5)	243/338 (72)	25 ^c^
Complete occlusion	253/290 (87)	25 ^d^
Complications	87/356 (24)	26
Leading to a permanent neurological deficit in N patients	32/265 (12)	21 ^e^
Rebleeding	11/368 (3)	26
All-cause mortality	50/357 (14)	26

GOS = Glasgow Outcome Scale; mRS = modified Rankin Scale; N = number. ^a^ Without study of Yang et al. (2017). ^b^ Without studies of Mokin et al. (2018) and Chalouhi et al. (2015). ^c^ Without study of Manning et al. (2019) due to no follow-up after discharge. Based on the available data of included studies. (Data were only missing for 5 patients in the study of Mokin et al. (2018).) ^d^ Based on the available data of survivors of included studies. The study of Manning et al. (2019) was not included due to a follow-up of only ~7 days and combining RR1 + 2. -When aneurysms of survivors with a missing follow-up are considered as non-occluded, then a complete occlusion: 225/280 (80%) → the studies of Manning et al. (2019) and Mokin et al. (2018) were not included. ^e^ Without (since unknown) the studies of Aguilar-Perez et al. (2020), Baker et al. (2020), Chan et al. (2014), Kaschner et al. (2019), and Mokin et al. (2018).

**Table 2 brainsci-12-00394-t002:** Logistic regression model for the favorable mRS after treatment with FD.

Predictor Variable	OR (95% CI)	*p*-Value
Unfavorable presentation (WFNS and HH 4–5)	0.156 (0.064–0.382)	<0.01
Saccular aneurysm	2.142 (0.781–5.870)	0.14
Aneurysm location (posterior circulation)	0.763 (0.331–1.763)	0.53
Aneurysm size (in mm)	0.883 (0.826–0.944)	<0.01
Treatment delay (in days)	1.053 (0.945–1.174)	0.35

CI = confidence interval; FD = flow diverter; HH = Hunt and Hess grading system; mRS = modified Rankin Scale; OR = odds ratio; WFNS = World Federation of Neurosurgical Societies grading system. The regression formula (after shrinkage) is: ln(*p*(favorable mRS)/(1 − *p*(favorable mRS))) = 1.695 + ((−1.6535 × unfavorable presentation (0 = WFNS and HH 1–3, 1 = WFNS and HH 4–5)) + 0.6788 × saccular aneurysm (0 = no, 1 = yes) + 0.2408 × aneurysm location (0 = anterior circulation, 1 = posterior circulation) + (−0.1106 × aneurysm size (in mm)) + 0.0462 × treatment delay (in days)).

## Data Availability

Data are available upon reasonable request from the corresponding author.

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
