# Peer review of "Outcomes after Flow Diverter Treatment in Subarachnoid Hemorrhage: A Meta-Analysis and Development of a Clinical Prediction Model (OUTFLOW)"

_brainsci, 2022, doi:10.3390/brainsci12030394_

Round 1
Reviewer 1 Report
I think this paper is useful as a prelude to future papers regarding the use of flow diverter treatment in aSAH, particularly with the advent of new devices (e.g Pipeline Shield).
On that note, it'll be helpful if the authors specified the devices used in the studies they cited, also if available, the number of stents used, as well as the PRU and the corresponding antiplatelet regimen in the peri-procedural period (if the data is available)
Author Response
Dear Reviewer,
Please see the attachment below.
Kind regards,
Viktoria Shimanskaya

Reviewer 2 Report
The authors provided a detailed and comprehensive meta-analysis of the role of FD in SAH. The article is well structured, well tracked, I just want to make a few suggestions.
- The Authors made a comparison between the conventional method and the methods performed with FD for rebleeding rate at row 264-265. It would be worthwhile and useful to consider comparing FD and conventional methods for all-cause mortality, favorable outcome, and occlusion rate. This will help the reader to place the use of FD in the treatment of acute SAH.
- Is there any information in the studies studied on what types of devices were used to perform the interventions? (Silk, Fred, p64, Surpass,..)
- The meta-analysis reviews 10 years of research. During this time, significant technical progress has been made in the quality of FDs. It would be advisable to indicate the type of device used in the studies, even in the form of a supplementary table.
Author Response

(The authors gave the same response as above.)

Round 2
Reviewer 2 Report
The authors have made the changes I have requested, so the manuscript is suitable for publication.